# A Comparison of the Response of the Human Intestinal Microbiota to Probiotic and Nutritional Interventions In Vitro and In Vivo—A Case Study

**DOI:** 10.3390/nu17193093

**Published:** 2025-09-29

**Authors:** Agnieszka Rudzka, Ondřej Patloka, Magdalena Płecha, Marek Zborowski, Tomasz Królikowski, Michał Oczkowski, Danuta Kołożyn-Krajewska, Marcin Kruk, Marcelina Karbowiak, Wioletta Mosiej, Dorota Zielińska

**Affiliations:** 1Department of Dietetics and Food Studies, Faculty of Science and Technology, Jan Dlugosz University in Czestochowa, Al. Armii Krajowej 13/15, 42-200 Częstochowa, Poland; d.kolozyn-krajewska@ujd.edu.pl; 2Department of Food Technology, Faculty of AgriSciences, Mendel University in Brno, Zemědělská 1, 61300 Brno, Czech Republic; xpatlok3@mendelu.cz; 3Institute of Biochemistry and Biophysics, Polish Academy of Sciences, Pawińskiego 5a, 02-106 Warszawa, Poland; mk.plecha.ibb@gmail.com; 4The Faculty of Medicine and Health Sciences, University of Applied Sciences in Nowy Sącz, Kościuszki 2G, 33-300 Nowy Sącz, Poland; mzborowski@ans-ns.edu.pl; 5Department of Dietetics, Institute of Human Nutrition Sciences, Warsaw University of Life Sciences (WULS-SGGW), Nowoursynowska 159 C, 02-776 Warsaw, Poland; tomasz_krolikowski@sggw.edu.pl (T.K.); michal_oczkowski@sggw.edu.pl (M.O.); 6Department of Gastronomic Technology and Food Hygiene, Institute of Human Nutrition Sciences, Warsaw University of Life Sciences (WULS-SGGW), Nowoursynowska 159 C, 02-776 Warsaw, Poland; marcin_kruk@sggw.edu.pl (M.K.); wioletta_mosiej@sggw.edu.pl (W.M.)

**Keywords:** microbiota, in vivo vs. in vitro conditions, probiotics, dietary macronutrients

## Abstract

**Background/Objectives:** With increasing knowledge of the role of the microbiota in health and disease, the need for the reliable simulation of its behavior in response to various factors, such as diet and probiotic administration in in vitro conditions, has emerged. Although many studies utilize developed systems, data on how accurately these systems represent individual microbiota responses are scarce. **Methods:** In the present study, the Simulator of Human Intestinal Microbial Ecosystem (SHIME^®^) was exposed to experimental conditions mimicking the application of probiotics and dietary changes in the study participant. Next-generation 16S rRNA sequencing was used to reveal the structure of the microbial communities in the analyzed samples. **Results:** Analysis of 17 samples revealed that predominantly diet and, to a lesser extent, probiotics had a divergent effect on the microbiota’s fluctuations dependent on the culture environment. Despite this, results from both in vitro and in vivo conditions aligned well with previously published data on the expected impact of dietary changes on the intestinal microbial community. **Conclusions:** The anecdotal evidence presented in this study suggested that current in vitro technology enables the reproduction of some of the microbiota responses that are well known from in vivo research. However, further work is required to enable simulations of an individual microbiota.

## 1. Introduction

The human gut microbiota plays a crucial role in maintaining the host’s health. It has been shown to influence processes such as immune modulation, metabolism, nutrient absorption, and protection against pathogens [1,2,3]. Disruptions in its composition, known as dysbiosis, have been associated with numerous diseases, including obesity, type II diabetes, inflammatory bowel disease, and neurological disorders [4,5]. The growing recognition of the microbiota’s role has sparked interest in how various interventions, such as dietary changes or probiotic supplementation, can be used to shape its composition and function.

Due to ethical, logistical, and biological constraints in human studies, in vitro gut models have become valuable tools for microbiota research, providing physiologically relevant insights into food–microbiota interactions and serving as important complements to in vivo studies, allowing for mechanistic validation under controlled conditions [6]. One of the most advanced systems is the Simulator of the Human Intestinal Microbial Ecosystem (SHIME^®^, referred to as SHIME through the text), which enables controlled, reproducible simulations of the human gut environment, including compartmentalized sections of the gastrointestinal tract (GIT) [7]. Recent research has increasingly focused on identifying in vitro models that closely replicate in vivo conditions, and SHIME has shown promise as a valuable predictive model to investigate microbial responses and interactions, particularly in the fields of food and nutrition, pharmacology, and gut health [8].

Although advanced in vitro models are widely used, data on how precisely they mirror individual in vivo conditions remain limited. Few works using the SHIME model have demonstrated qualitative similarities to microbiota shifts reported in human studies following similar interventions [9]. Nevertheless, several knowledge gaps can be pointed out. Most studies utilize SHIME and compare its output with findings from the literature [10,11,12]. Whereas fewer investigations are pointed towards directly comparing these changes to the microbiota of the donor, whose feces were used to inoculate the SHIME system under matched interventions [13]. Several studies have demonstrated that the composition of the original microbiota is not preserved after the transfer to the simulator [13,14,15]. This raises questions about whether such microbiota retain the ability to respond to a stimulus in a similar way as they would in their natural environment.

Access to in vitro tools that allow reliable simulations of an individual microbiota’s behavior could support advances in personalized health management, including precision nutrition. The term “precision nutrition” was introduced in the early 2010s and reflects a shift toward integrating omics data—genomic, metabolomic, and microbiomics—to design interventions that maximize individual health benefits [16,17]. In this context, the gut microbiota is both a target and a mediator of personalized dietary strategies. In vitro models offer a valuable platform to assess microbiota shifts and estimate diet-induced microbial metabolite production [18]. However, currently, data are insufficient to learn how a pool of macronutrient-level dietary changes, rather than isolated compounds like prebiotics or polyphenols, influences the microbiota composition in the SHIME relative to in vivo conditions. Bridging this gap would enhance the validity of in vitro systems for simulating complex host–microbiota–diet interactions.

Besides diet, a known approach to modulate the microbiota is the administration of probiotics. *Lacticaseibacillus rhamnosus* GG (LGG) is one of the most extensively studied probiotic strains, known for its beneficial effects on gut barrier function, immune response, and microbial balance. Studies have shown that LGG can temporarily colonize the GIT and promote shifts in microbial compositions, often characterized by increases in beneficial taxa and metabolic outputs such as short-chain fatty acids [19,20].

In this study, dietary and LGG interventions were applied to the SHIME system and the human donor in parallel to discern whether the behavior of individual microbiota may be modelled in an in vitro setting.

To our knowledge, this is the first study that attempted a comparison of the microbiota response to dietary and probiotic interventions in humans and simulators in parallel.

## 2. Materials and Methods

### 2.1. Study Design

The design of this study and methodological details were previously described by Rudzka et al. (2023) [13]. This previous publication reported a subset of data from the same experiment; hence, to avoid repetition, methodological details contained in the current work were limited to a minimum.

Briefly, a 39-year-old female volunteer with mild hypercholesterolemia but who was otherwise healthy donated a sample of feces that was used to inoculate the SHIME system (ProDigest, Gent, Belgium). The system operated in a MultiSHIME mode, focusing on the distal colon (two replicates of the following setup: one stomach, one duodenum, one proximal colon, and three distal colon bioreactors). Before the experiment, SHIME was stabilized for 14 days, using the standard SHIME media as per the manufacturer’s protocol. The volunteer provided 28-day dietary records before the start of the experiment. The experiment, during which the interventions were administered, lasted 28 days.

### 2.2. Interventions

Both the SHIME model and the volunteer were subjected to parallel interventions: a 28-day dietary modification and supplementation with a commercial formulation containing LGG (ATCC 53103) administered during the first 14 days of dietary intervention.

A commercially available LGG supplement in the form of capsules, claimed to contain 6 × 10^9^ LGG cells per capsule (other ingredients and additives: maltodextrin, hydroxypropyl methylcellulose, magnesium salts of fatty acids), was used in this study. Both the volunteer and the SHIME ingested the supplement twice a day. The volunteer consumed the capsules, while SHIME was inoculated with a small volume of sterile water in which the contents of 3 capsules were suspended (1.5 capsules in each stomach/ileum vessel). SHIME’s stomach was administered LGG right after filling with the fresh feed medium at pH 2, thereby exposing the probiotic to simulated GIT conditions from the stomach to the colon.

The volunteer was fed a nutritionally balanced diet with a calculated content of dietary macronutrients. To match macronutrient changes in the volunteer’s diet, the standard SHIME nutrient medium was modified by adjusting the proportions of its components that simulated dietary residues of animal protein (special peptone), non-animal protein (yeast extract), fiber (xylan, arabinogalactan, pectin, starch), and sugars (glucose). Fat was not included, as in this study, we chose to modify the standard SHIME feed by an exclusive adjustment of nutrient proportions. Nevertheless, the diet of the volunteer was not fat-free.

The mean intakes of dietary macronutrients and the nutritional compositions of SHIME feed media before and during the intervention are summarized in Table 1. Daily detailed nutrition data, including information about the meals given to the volunteer, were deposited in an open data resource [21,22].

The modifications of the standard SHIME feed media were guided by an estimated proportion of food residues that were expected to reach the colon of the volunteer. This estimate was derived from a study on ileostomised humans, where the authors found that on average, 83 ± 15% of fiber, 1.1 ± 0.3% of carbohydrates, and 16 ± 5% of protein that were ingested reached the ileostomies [23]. These residue fractions were used to calculate nutrient concentrations that were included in the SHIME media while applying a normalization to soluble fiber content. To calculate SHIME medium composition, the following nutrient-to-ingredient conversion factors were used—1:1 for animal protein to special peptone, resistant starch to starch, arabinogalactan + arabinoxylan to sum of xylan and gum arabic, pectin to pectin, and 1 to 1.54 for non-animal protein to yeast extract. The glucose content of the media was corrected for the presence of glucose in yeast extract (2.9%).

### 2.3. Sampling

Twelve samples of feces (samples denoted with letter K and I-inoculum) and 11 samples of liquid from SHIME’s distal colon compartments (samples denoted with letter L) were collected. The sampling scheme is depicted in Figure 1. In total, three samples were collected before the experiment (I; and directly before the application of the intervention: SK and SL). Whereas during the intervention, sampling was performed 10 times, of which 5 samples were collected during the combined dietary and probiotic interventions (on experimental days 3, 6, 8, 10, 12) and the remaining during probiotic washout and continued dietary intervention (on experimental days 16, 18, 22, 25, 28). The sampling scheme relied on the volunteer’s availability to provide a fresh fecal sample on-site. The volunteer was instructed on the required number of samples for each study stage, with the condition that successive samples be collected at least one day apart. Samples from SHIME were collected in parallel with fecal samples provided by the volunteer.

### 2.4. Sample Preparation and Analysis

Methodological details of sample preparation and analysis were described previously [19]. Briefly, the DNA was extracted in three replicates from each fecal sample and from three of SHIME’s distal colon compartments. Then, the quality of the DNA was confirmed spectroscopically and via polymerase chain reaction. To reveal the structure of microbial community, a high-throughput sequencing of the V3–V4 regions of 16S rRNA was performed using DNA extracts pooled from three replicates taken for each sample. In total, 23 samples described in this study were sequenced.

### 2.5. Statistical Analysis

The statistical analysis presented in the current publication was performed with the use of three types of software. Statistica ver. 13 (StatSoft, Kraków, Poland) was used to obtain Spearman’s correlation coefficients for sensitivity analysis and run Wilcoxon signed-rank test. R built into R studio (Posit, Boston, MA, USA) was used to 1. calculate and visualize the microbial diversity indices and obtain the Jaccard distance matrix for the principal coordinate analysis (PCoA), 2. calculate data for heatmaps and draw them, and 3. perform linear modeling as the main analysis to compare the response of the microbiota to interventions in two different environments and draw a biplot. The linear modeling was preceded by data transformation. The abundance of microbial genera/phyla was centered log-ratio transformed (CLR) [21]. Dietary macronutrient residues expressed as g/day (volunteers) or g/L (SHIME) were standardized within each environment separately using z-transformation. Then, the principal component analysis (PCA) was performed on both CLR-transformed microbial counts and standardized macronutrient content, and the first two principal components (PCs) for each analysis were retained. These PCs were then used in the linear model described by relationship (1):PC1_micro or PC2_micro ~ (PC1_diet + PC2_diet + Probiotic) × Environment(1)
where

PC1_micro and PC2_micro—principal components of microbial community abundance.

PC1_diet and PC2_diet—principal components of the concentration of macronutrient residues in both in vitro and in vivo settings.

Probiotic—presence/absence of LGG supplementation.

Environment—in vitro (SHIME) or in vivo (volunteer) microbiota cultivation setting.

Linear models were evaluated by means of Type III Analysis of Variance (ANOVA) to test main and interaction effects.

Microsoft Excel (Microsoft, Redmond, WA, USA) was used to obtain the remaining figures.

All statistical tests assumed a significance level of 0.05. For multiple comparisons, either Benjamini–Hochberg false discovery rate (FDR) or Bonferroni corrections were used, as specified in the results. Compositional data of microbiota abundance were CLR transformed for correlation analyses.

## 3. Results

### 3.1. The Composition of the Microbiota

Substantial differences in the microbiota composition were observed between fecal and SHIME samples on all taxonomical levels, from phylum to genus. The dissimilarities in the microbial composition between the SHIME and stool microbiota were clearly visualized on the PCoA plot (Figure 2), where distinct groupings for each of the settings can be noticed. However, the plot also suggested the existence of some outliers. Three outlying samples were identified in each of the stool and SHIME datasets. These samples included L3, L4, and L10 for SHIME and K3, K4, and K10 for the stool. Consequently, they were excluded from the further statistical analyses presented in this manuscript.

The bioinformatic analysis identified 112 bacterial genera in the whole dataset from the study; however, only 20 and 25 of these genera appeared in the majority of the analyzed fecal (at least 9 out of 12) and SHIME (at least 8 of 11) samples, respectively. The abundance of the most prevalent genera in all samples (including outliers) is shown in Figure 3. Eight of these genera were shared between the SHIME and stool, including *Bacteroides*, *Bifidobacterium*, *Akkermansia*, *Alistipes*, *Sutterella*, *Parabacteroides*, *Phascolarctobacterium*, and *Victivallis*. Their abundance varied between in vivo and in vitro settings and fluctuated over time. Among these genera, only *Bacteroides* was observed in appreciable levels in all the samples. Notably, the SHIME microbiota was richer in *Bifidobacterium*, *Akkermansia*, *Lachnoclostridium,* and *Fusobacterium*, while *Alistipes*, *Subdoligogranulum*, Oscillospiraceae (UCG-002), and members of the Christensenllaceae (R-7 group) were more abundant in fecal samples.

Overall, the data shown in Figure 3 suggest that the microbial community was undergoing dynamic changes during the experiment in both in vitro and in vivo conditions, without any time-dependent trends.

Similarly to the taxonomic abundance, the Firmicutes to Bacteroidetes (F/B) ratio fluctuated during the experiment in both in vivo and in vitro conditions. This fluctuation was independent of the experimental phase, and its patterns were dissimilar between the SHIME and feces (Appendix A). Overall, the F/B ratio was greater in fecal (mean: 4.9) than in SHIME samples (mean: 0.7). Except for outliers, neither of the fecal samples was characterized by an F/B ratio below one, but all of the SHIME samples fell below this threshold.

### 3.2. Microbiota Genetic Diversity

The α-diversity: The Chao1, Shannon, and Simpson indices of the microbial communities differed between fecal samples from the study participant and those from the SHIME model (Figure 4).

Fecal samples were characterized with consistently higher operational taxonomic unit (OTU) richness regardless of the α-diversity expression. This difference was statistically significant after Wilcoxon’s signed-rank test (*p* < 0.05) only for the Chao1 index; however, it did not survive the Bonferroni correction.

Twenty of the genera detected in the inoculum were absent from all SHIME samples. These genera were, however, each present at <4% abundance within the inoculum. On the other hand, seven genera were specific to SHIME samples, albeit also at a very low abundance (<2%). These genera included *Enterococcus*, *Clostridium sensu stricto* 1, *Eubacterium hallii* group, *Megasphaera*, *Hafnia-Obesumbacterium*, *Bacillus*, and *Stenotrophomonas*, which are all known members of intestinal microbiota.

### 3.3. The Comparison of the Microbiota Response to the Probiotic and Dietary Intervention

In previous sections, differences between the microbiota composition of a donor in the original in vivo conditions and after establishment in the in vitro conditions were presented. However, the main research question of this study was whether the microbiota behaved in a similar fashion regardless of the environment in which it was cultured.

During the probiotic supplementation, certain shifts in the abundance of the most prevalent genera were noticed (Figure 3). In fecal samples, the relative abundances of *Phascolarcobacterium* and Christensenellaceae (R-7 group) increased, while *Bacteroides* showed a consistent decline throughout the intervention, from the baseline (sample SK) through K1 and K2 to K5. On the other hand, SHIME samples exhibited a consistent increase only in the relative abundances of *Sutterella* and *Victivallis*.

Interestingly, the changes in the abundance of bacterial genera observed during the probiotic intervention did not reverse after its cessation. Furthermore, some shifts in the microbiota were noted during the post-supplementation probiotic washout period. These included a steady increase in *Akkermansia* and a decrease in *Barnesiella* in the volunteer’s microbiota. In the SHIME system, this period was marked by a decline in *Bacteroides* and increases in *Oscillibacter* and *Bilophila*.

Although an increase in the relative abundance of the former group of *Lactobacillus* (as classed under the older nomenclature and identified using the SILVA database for OTU assignment) was expected during the probiotic intervention as well as a decrease after its cessation, only in SHIME was such a phenomenon observed. The relative abundance of *Lactobacillus* was at 0.5% before the supplementation period (sample SL) and ranged from 2.23 to 2.63% while it lasted (samples L1, 2, and 5) and ranged from 1.21 to 1.89% after its cessation (samples L6, 7, 8, and 9). *Lactobacillus* was detected in all SHIME samples. On the other hand, in feces, this genus was only present in the sample used for the inoculum preparation at a relative abundance of 2.2%. After that, it was not detected in any of the samples collected during or after the probiotic intervention.

The dietary intervention did not seem to have an immediate effect on the microbiota structure in either of the two culturing environments. Correlation analysis between the relative abundance of microbial genera and macronutrient concentrations—measured either in food residues from the volunteers’ diet or in SHIME media from the day before sampling—revealed that only four of the most prevalent genera in each microbiota culturing environment increased in abundance in response to higher concentrations of at least one of eleven macronutrients (Spearman’s correlation coefficient > 0.7). To explore potential cumulative effects, a sensitivity analysis with a moving average for macronutrient residues was performed (results—Figure 5, full dataset available in data repository [22]). The analysis suggested that the optimal averaging window (the greatest number of genera with abundance correlating positively with at least one macronutrient) was 9 days for the volunteer and 12 days for SHIME.

Although the microbiota is known to respond quickly to dietary changes (shifts are noticeable within 24 h [24]), validation studies carried out in SHIME showed that these responses stabilize after a certain period (12 days for a standard, unchanged nutritional medium [25]). This could explain why a low number of correlations was found with the 24 h data in the present study and prompted the use of an optimal macronutrient content averaging window in all further analyses.

To quantitatively assess whether the probiotic and dietary intervention had different outcomes in two microbiota culturing environments, a linear model was fitted to bacterial abundance and intervention data (Section 2.5). The analysis focused on eight genera prevalent in the majority of SHIME and fecal samples (Section 3.1) and four selected macronutrients (soluble fiber, sugars, non-animal protein, and animal protein). A parallel analysis, at a phylum level, confirmed that the significance of the effects translated to the whole studied microbial population (open data [22]). The type III ANOVA analysis did not detect significant main effects for the intervention variables across the pooled dataset for eight selected genera (SHIME and volunteer considered together). In contrast, a strong main effect of the Environment on PC1_micro (explaining ~62% variance) was noted. Furthermore, a significant interaction between the Environment and PC1_diet and Environment and Probiotic on PC1_micro (~7% and 4% variance explained, respectively) was identified. Out of these results, at the phylum level, only the interaction effect between the Environment and Probiotic was not significant, whereas a significant main effect of PC1_diet and its interaction with the Environment on PC2_micro was found.

In summary, the structure of microbial communities was significantly dependent on the culturing environment (volunteer vs. SHIME), and the microbiota responded differently to the dietary intervention in each environment (regardless of whether the whole population or a small subset of eight genera common for both SHIME and the volunteer was considered). The biplot (Figure 6) illustrated this divergence well. Fecal and SHIME samples were clearly grouped in separate clusters, with an overlap of 95% confidence areas (overlap not observed at a phylum level [22]). In general, the SHIME microbiota was characterized with a greater abundance of the most prevalent genera shared between the two environments, and samples taken during the probiotic intervention presented an even greater enrichment. The only exception was *Alistipes*, which seemed to be more abundant in fecal microbiota.

The biplot included overlaid vectors from a PCA on macronutrient concentrations. The abundance of most genera selected for analysis showed a positive correlation with at least one macronutrient, either animal protein, non-animal protein, or soluble fiber. Since the microbiota composition was significantly affected by the interaction between the Environment and PC1_diet, a separate analysis investigating responses to macronutrients in each environment was performed. For this purpose, a series of Spearman’s correlation analyses was conducted (Figure 7). Besides the eight genera highly prevalent in SHIME and fecal samples, broader descriptors of the microbiota’s condition, such as α-diversity indices and the F/B ratio, were also included.

Sperman’s correlation analysis revealed that before FDR corrections, there were some statistically significant relationships between macronutrients and microbiota across all analyzed levels. After applying the FDR adjustment, a significant increase in the response to macronutrients (non-animal protein and soluble fiber) was found only for the F/B ratio in the stool. Other positive correlations that were initially significant but did not survive the correction displayed a strong relationship (Spearman’s ρ > 0.7). Such relationships were identified on the genus level but not for α-diversity indices. In stool, the abundance of *Akkermansia* correlated positively with non-animal protein; *Phascolarcobacterium* with soluble fiber; and *Victivallis* with the animal protein content in the dietary residues. In contrast, in SHIME the abundance of *Akkermansia* correlated negatively with the non-animal protein content, which remained significant even after the FDR correction. The other two genera showed a positive but weak association with their respective macronutrients.

## 4. Discussion

There are not many studies evaluating the fate of the microbiota of a particular donor in SHIME and its natural environment by means of modern high-throughput DNA sequencing techniques. Early validation studies (from the end of the XX and beginning of the XXI century) used such indicators as the metabolism of the microbiota, plate counts of selected taxa, and the stability of the microbiota after inoculation [13]. Outcomes were typically compared with data from existing human trials that did not include fecal sample donors; however, these comparisons allowed the authors of previous studies to demonstrate that the system was able to mimic the behavior of human microbiota. Multiple modern studies, which also relied on these non-parallel comparisons, demonstrated that high-throughput DNA sequencing techniques allowed for the validation of the outcomes from SHIME, e.g., [9,26]. Thus, it is not surprising that in the modern literature, SHIME has established itself as a state-of-the-art, validated method for studying reactions of the microbiota to various interventions in in vitro conditions.

In this study, it was demonstrated that the changes in the microbiota composition in response to applied interventions varied noticeably between in vitro and in vivo conditions. Hence, to validate the present findings, we followed an approach driven by previous records and evaluated how the responses of microbiota to probiotic and dietary interventions reported in this study aligned with the existing scientific evidence.

The former group of *Lactobacillus* was not detected in fecal samples during and after the LGG supplementation. This was unexpected, as LGG is well-characterized, and it has been known to appear in feces not only during but also up to 12 days post-supplementation [27]. On the other hand, in the SHIME model the former group of *Lactobacillus* persisted through the whole experiment, during and after the probiotic supplementation. This observation aligns well with both in vivo and in vitro studies. The latter showed that the system may allow LGG proliferation in the ascending colon for at least 10 days and promotes its persistence in the distal colon for at least two weeks [26,28].

In contrast to the human gastrointestinal tract, the SHIME model provides stable and controlled conditions that facilitate the persistence and proliferation of supplemented strains, whereas in vivo their colonization is often limited by microbial competition, rapid intestinal transit, and host immune responses. Moreover, colonization by *Lactobacillus* is frequently transient, meaning that their presence in feces may be sporadic and below the detection threshold of 16S rRNA sequencing. In addition, the 16S rRNA may suffer from a primer-dependent amplification bias towards low-abundance DNA, which ultimately can result in false negatives. Moreover, LGG may transiently adhere to the mucosal surface of the small intestine or proximal colon rather than being shed in feces, which limits its detectability in stool-based sequencing.

Besides the increase in the former *Lactobacillus* genus abundance, the literature reported some other minor effects of LGG supplementation on the microbiota. Taxa that were previously shown to respond to such supplementation in human and animal studies reported increases in Lachnospiraceae and Porphyromonadaceae families [29] and genera such as *Bacteroides*, *Ruminococcus*, *Butyricicoccus*, *Erysipelatoclostridium*, *Flavonifractor*, and *Bacillus* [30,31]). In our study these taxa were present at very low abundances, except for *Bacteroides*, which tended to decrease during the probiotic intervention in the volunteer’s feces. Furthermore, in vitro research with the use of the SHIME model inoculated with a western-diet consuming donor’s microbiota reported a minor decrease in the F/B ratio after 6 days from a single introduction of LGG into the system [28]. In contrast, in this study, the F/B ratio increased during the intervention in both in vitro and in vivo conditions and fluctuated after its cessation (Appendix A). These slight changes and lack of clear trends in their reversal during the LGG washout period confirmed that dietary intervention was a major factor driving the fluctuations in the structure of the microbial community in this study.

The literature indicates a positive relationship between the diversity of the microbiota and dietary content of animal protein, plant protein (particularly soy), and fiber—especially branched arabinoxylan, fermented foods, and overall dietary variety [32,33,34,35]. On the other hand, diets high in fat and sugar, as well as the excessive consumption of ultra-processed foods, are known to reduce bacterial diversity [35,36]. This is consistent with the broader understanding that the diminished diversity of the microbiota is detrimental to host health and is associated with diseases, including diet-related conditions such as obesity or type 2 diabetes [37,38,39].

This study did not find any strong positive associations between macronutrients and microbiota’s α-diversity in either in vitro or in vivo conditions (Figure 7). On the other hand, a strong (but not statistically significant) negative relationship between the content of non-animal protein seemed to decrease the diversity of fecal microbiota. Such negative associations should be interpreted with care. Macronutrients do not directly contribute to the reduced ability of specific microorganisms to thrive. Rather, they may promote competing microorganisms, suppressing the ones that cannot readily utilize particular energy sources. Therefore, a decline of α-diversity resulting from the loss of the abundance of specific groups of microorganisms in response to a nutrient is likely a result of coadaptation issues. These challenges are particularly relevant to in vitro systems, such as SHIME, where the microbiota composition is affected during the transfer from in vivo conditions [14,15].

Apart from changes in diversity, research often refers to the F/B ratio to assess the condition of the microbiota and its reaction to a particular intervention. In this study, higher levels of non-animal protein and soluble fiber promoted Firmicutes over Bacteroidetes (Figure 7) but only in fecal samples. These findings align well with the literature reports, where an increase in the abundance of Firmicutes was reported in response to whole grain consumption [33] and in individuals adhering to a healthy diet pattern [40]. Furthermore, a decrease in Firmicutes after a few days (ca. 5) of an animal-based diet has been reported [41].

Although the elevated F/B ratio is an established indicator of gut microbiota dysbiosis and has been associated with diseases, such as obesity, in multiple studies [42,43], it is important to note that many beneficial microbes, including the former genus *Lactobacillus*, belong to the Firmicutes. In this study, SHIME and fecal samples contained an appreciable proportion of Firmicutes composed predominantly of three families: Lachnospiraceae, Ruminococcaceae, and Monoglobaceae. These families are known to feed on dietary fiber, with Lachnospiraceae and Ruminococcaceae being well-established short-chain fatty acid (SCFA) producers [44]. A high abundance of Lachnospiraceae and Ruminococcaceae has been associated with good health and favorable dietary habits, i.e., adherence to the Mediterranean diet [45,46]. In addition, such a microbiota is characteristic of vegetarian and vegan over omnivorous feeding patterns [47].

Considering a deeper taxonomic level, this study found strong positive associations between the relative abundance of *Akkermansia*, *Phascolarcobacterium*, *Victicvallis*, and *Bifidobacterium* and the concentration of some of the macronutrients in either in vitro or in vivo conditions (Figure 7).

A very well-known *Akkermansia* species is *Akkermansia muciniphila*. This species is related to healthy metabolism [48] and is promoted by various dietary fibers, including arabinogalactan, arabinoxylan, resistant starch, and pectin [49]. Interestingly, even fiber-restricted diets, such as the low-FODMAP [50] and ketogenic [51] diets, were shown to increase its abundance in vivo. Conversely, high-animal-protein diets could diminish the population of *Akkermansia* [52], while dietary sugars may promote their abundance [53]. Animal studies point to a positive effect of unsaturated fat consumption on the levels of *A. muciniphila* [32], a finding partly supported by a human study where people with mild cognitive decline followed a Mediterranean–ketogenic diet and also experienced an increase in *Akkermansia* [54]. Our results appear to support the role of plant-based diets in promoting *Akkermansia.* We have found a strong, positive correlation between its abundance and the concentration of non-animal protein in food residues in vivo. Notably, this association was not replicated in vitro; in fact, an inverse relationship was observed in SHIME. This discrepancy could result from non-animal protein being represented in SHIME feed by the yeast extract.

In our study, the abundance of *Phascolarcobacterium*, associated with good metabolic health [55], showed a strong positive correlation with the content of soluble fiber in the residues from the volunteers’ diet (Figure 7), confirming the literature findings [56,57,58,59]. However, this effect was not replicated in SHIME. Moreover, the abundance of *Victivallis* was positively correlated with the content of animal protein in the residues within the volunteer’s diet and pectin in SHIME (Figure 7), which was also in agreement with the existing literature [53,59,60,61]. The role of *Victivallis* in human health is not yet fully understood. Its abundance was found to be positively associated with the increased risk of hypertension and radiotherapy-induced oral mucositis in patients with head and neck cancer [62,63]. On the other hand, this genus belongs to SCFA producers and has been shown to increase in abundance in response to a reduced-sucrose and starch diet, alleviating symptoms in patients with irritable bowel syndrome [62,64].

The current study also found that increased levels of glucose promoted *Bifidobacterium* in SHIME (strong, but not significant correlation) but not in fecal samples (Figure 7). *Bifidobacterium* is a broadly recognized probiotic genus with anti-inflammatory and pathogen-protective effects [65,66]. It ferments a broad range of carbohydrates, including digestible and indigestible fractions, producing beneficial SCFAs [32,67]. Unsurprisingly, *Bifidobacterium* is promoted by carbohydrate-rich foods and supplements, such as corn flakes, apple pomace, inulin, β-glucan, or type 2 and 4 resistant starch [32,68,69,70]. Murine studies have shown that a diet high in unsaturated fats also supports this genus [32]. In contrast, low-carbohydrate and high-fat feeding patterns contribute to the decrease in *Bifidobacterium* [54,71].

Overall, observed associations between the macronutrient concentration and the abundance of specific taxa were all in good agreement with previously published findings from human and animal studies. This suggests that although we failed to reproduce the response of the studied individual microbiota to interventions in in vitro conditions, the SHIME system still provided outcomes that reflected well-known microbial responses to dietary inputs from human and animal studies.

One of the multiple reasons for the failure to reproduce the behavior of an individual microbiota in vitro could be a difference in its structure at the beginning of the experimental intervention period. Existing research underscores the variability of microbiota responses to identical interventions. For example, recently Devarakonda et al. [72] reported that the microbiota of different subjects exhibited diverse responses to the supplementation with type 2 and 4 resistant starch. The greatest interindividual variation was found in the abundance of *Ruminococcus bromii* and *Parabacteroides distasonis*, both known to react to this fiber fraction. Similarly, an earlier study by Salonen et al. [73] showed that the responsiveness of microbiota to dietary interventions in obese men was highly individualized. Furthermore, studies imply that the interactions of the intestinal microbial communities with medication are also specific to each person and can influence the efficacy of treatments [74]. These findings suggest that enabling in vitro research capable of evaluating the response of particular microbiota to nutritional or bioactive compounds could advance the development of targeted and effective interventions. In this context, the recreation of the microbiota of a particular donor in the SHIME and similar systems seems of great importance.

It is well understood that in vitro systems designed for human microbiota cultivation are imperfect in their reflection of realistic in vivo conditions. They simply lack the complexity of bodily functions that govern secretory activity, digestion, and absorption. The efficiency of these functions varies between individuals and is influenced by numerous factors such as diet, physical activity, sex, health status, mood, and many other personalized physiological responses. Such complexity of the individual GIT functions shapes the structure of the microbial community, and therefore, a shift in the microbiota composition upon transfer to in vitro conditions is expected.

However, the nature and extent of changes during the transfer of the microbiota to a simulator are not reproducible across different studies nor even among different donors within the same study [15]. For instance, here we did not detect a significant change in the α-diversity. This aligns with the findings of Liu et al. [14], who also studied fecal microbiota in SHIME. In contrast, another study reported a decrease in the Shannon index for the synthetic inoculum in the same system [75].

In the current study, the F/B ratio, crucial for distinguishing “healthy” and “dysbiotic” statuses, decreased approximately three-fold after the stabilization of the microbiota in vitro (Appendix A). Other studies that used the SHIME system with its standard startup protocol showed smaller changes in the F/B ratio (less than two-fold) and indicated that establishing a “dysbiotic” microbiota (with F/B ratio > 1) in SHIME was possible [14,15,42].

At deeper taxonomic levels, changes in the microbiota structure after the transfer to in vitro conditions were also reported [14,15,75], which we similarly observed. However, specific changes varied between studies, most likely due to differences in the structure and quality of the microbiota comprising the inoculum.

Beyond the inability to accurately reflect the complexity of the human GIT, the literature identified several particular factors that hamper the preservation of the microbiota’s structure during its establishment in the simulator. These factors include stochasticity, coadaptation, stressors acting upon the inoculum, and the feed media’s composition.

A study utilizing a synthetic inoculum, based on 23 species isolated from human feces and cultivated in a system of Multifors bioreactors, highlighted that stochasticity was the key driver of the microbiota structure during its establishment in vitro [76]. The authors failed to reproduce the structure of the microbial community between replicates of the same cultivation conditions.

Coadaptation issues were also reported, particularly if the members of microbiota from different donors were combined into a single inoculum [76]. While the mentioned study provides valuable insights into the dynamics of synthetic microbiota establishment in the bioreactor, further research is needed to understand how these findings translate to more complex inocula, such as the full fecal microbiota of a single person used here.

Research with the use of a synthetic inoculum showed that microbiota establishment in SHIME is strongly influenced by the feed medium composition. Differences in feed media accounted for approximately 38% of the observed variability in the microbial community structure [75]. In contrast, the proportions of the microorganisms within the inoculum seemed to have little effect on its structure after the stabilization process.

When using complex fecal microbiota, the medium composition seems to play a critical role not only during the experiment but perhaps predominantly during the transition into an in vitro setting. Marzoratti et al. [77] demonstrated that feed formulations enriched in either dietary fiber or protein led to distinct microbial communities in SHIME, even though the microbiota originated from the same donor. While the F/B ratio was similar for both conditions, the high-protein feed increased the level of Proteobacteria in SHIME [77].

These findings suggest that any efforts focused on reproducing the microbiota of the donor in SHIME should begin with carefully tailoring the feed composition, primarily during the stabilization period. In the present study, the SHIME feed was only matched with feed residues that could originate from the volunteer’s diet arbitrarily and only after the standard stabilization period. This likely contributed to adaptive shifts in the microbial community that may have limited its fidelity to the original human microbiota. Furthermore, the lack of some food residues, such as fat and insoluble fiber, in the standard (and our experimental) SHIME feed may contribute to these shifts. Moreover, other factors differentiating what ultimately reaches the microbiota in the colonic environment in vitro and in vivo cannot be neglected. For example, pancreatic juice used in the SHIME model (containing potent gut microbiota modulators such as bile acids) may induce excessive changes in the microbiota composition due to the absence of reabsorption and bacterial metabolites of excessive bile acids.

Besides the coadaptation and medium composition, a range of stressors could impact the establishment of the particular microbiota in the in vitro system. These include all the differences between the human organism and bioreactors, as well as the procedural steps involved in the inoculum preparation. The literature suggests that some of these stressors, such as frozen storage before inoculation, help to reproduce the microbiota structure in vitro at least at the phylum level [15]. Low temperatures can alter the gene expression of microorganisms, consequently making them less sensitive to other environmental changes [78,79]. As a result, freezing the inoculum before its introduction into the bioreactor may partially stabilize the microbial community and mitigate compositional shifts during the transition to in vitro conditions. Nevertheless, here we have chosen to work with a freshly prepared fecal inoculum (a standard approach), which may partly explain a large shift in the microbial community’s structure after stabilization in SHIME. 

In summary, findings of this and other studies suggest that standard protocols used for running in vitro experiments do not preserve the original structure of human microbiota. This likely contributes to shifts in the microbiota responses to interventions. Further studies should hence explore modifications to standard protocols aimed at the preservation of an individual microbiota structure. Such improvements could enable the use of in vitro technology to support developments in personalized health management strategies, including precision nutrition.

## 5. Conclusions

In this study, we found that the same initial microbiota exhibited different responses to probiotic supplementation and dietary changes in in vivo and in vitro settings. Thus, we failed to simulate the behavior of the individual microbiota in SHIME. Despite this, the system enabled microbiota responses that were known from multiple earlier human and animal studies. Particularly, we observed a positive correlation of the *Bifidobacterium* abundance with the increase in glucose in the feed medium and the enrichment of the former *Lactobacillus* genus during probiotic supplementation.

Although this study included no replication and only a single donor, to our knowledge, it presents the first attempt to compare the response of an individual microbiota to probiotic and dietary interventions in the donor and SHIME in parallel. We propose that the alteration of the microbial community during the establishment in the in vitro setting, shown here as well as in previous studies, is one of the several factors contributing to the observed divergence of the individual microbiota behavior between artificial and natural environments.

Nevertheless, the ability to replicate the reaction of an individual’s microbiota to a dietary, probiotic, drug, or any other intervention holds promise for the development of precision nutrition or personalized medical strategies. Further advances in the in vitro technology that would be aimed at the preservation of the structure of microbial communities contained in the inoculum, e.g., preparing frozen rather than fresh inocula and adjusting the feed media composition, are essential to enhance the predictive power of such systems.

## Figures and Tables

**Figure 1 nutrients-17-03093-f001:**
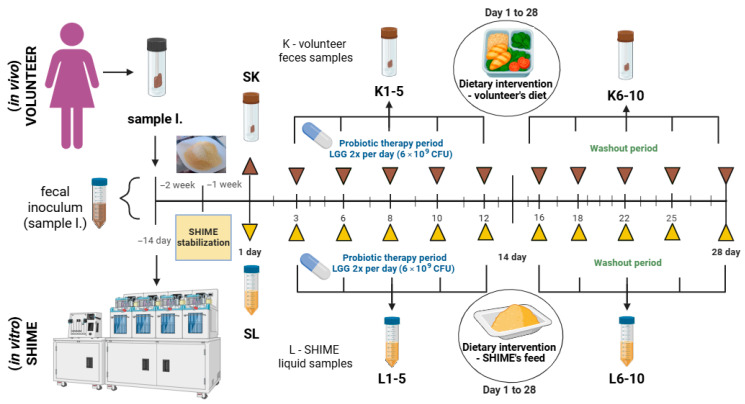
Experimental design and sampling scheme used in this study. Created with Biorender.com.

**Figure 2 nutrients-17-03093-f002:**
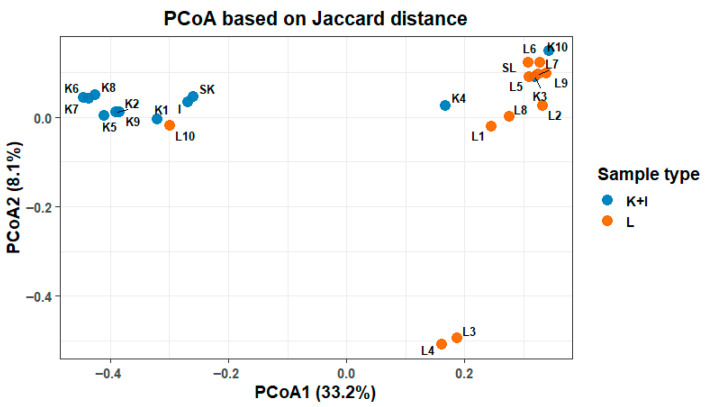
Principal coordinate analysis of the composition of microbiota in SHIME (L) and stool (K), including inoculum (I) samples.

**Figure 3 nutrients-17-03093-f003:**
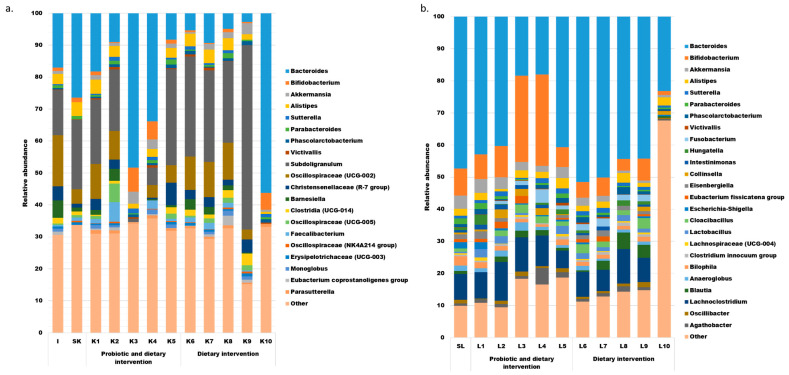
The most prevalent bacterial genera in the feces of the volunteer ((**a**), includes 20 genera detected in at least 9 samples) and in the liquid from SHIME ((**b**), includes 25 genera detected in at least 8 samples).

**Figure 4 nutrients-17-03093-f004:**
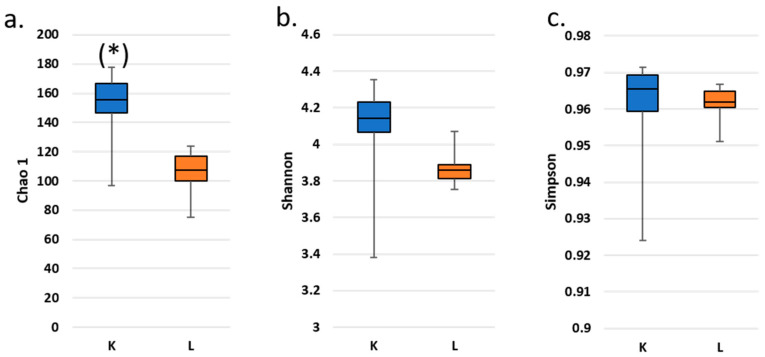
Box plots of fecal (K) and SHIME (L) microbial α-diversity: Chao1 (**a**), Shannon (**b**) and Simpson (**c**) indices along the experimental trial. Reported data exclude outliers and the inoculum. (*) denotes a statistically significant difference between α-diversity in K and L according to a Wilcoxon’s signed-rank test (*p* < 0.05). No significant differences between pairs of data were detected after Bonferroni correction.

**Figure 5 nutrients-17-03093-f005:**
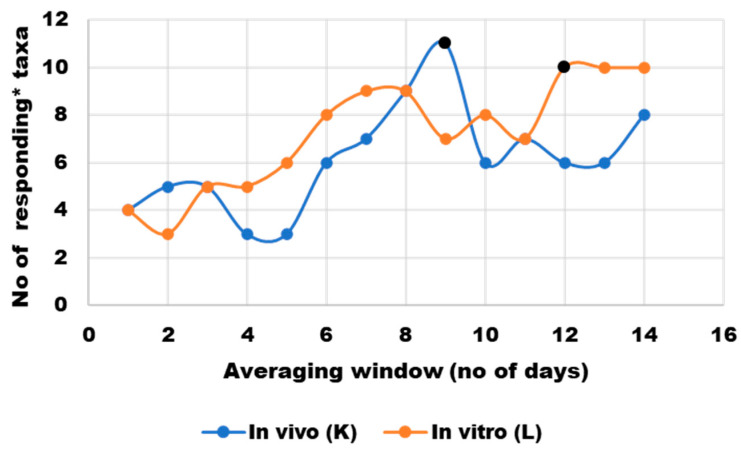
Sensitivity analysis of nutrient content averaging window to correct for a lag in microbial response to nutritional changes; * responding taxa were defined as those whose abundance correlated positively (Spearman’s ρ > 0.7) with at least one macronutrient or group of macronutrients (protein, animal protein, non-animal protein, arabinogalactan, arabinoxylan, arabinogalactan + arabinoxylan, pectin, soluble fiber, starch, sugars, protein-to-soluble-ratio). Black markers indicate averaging windows that were selected for further statistical analyses.

**Figure 6 nutrients-17-03093-f006:**
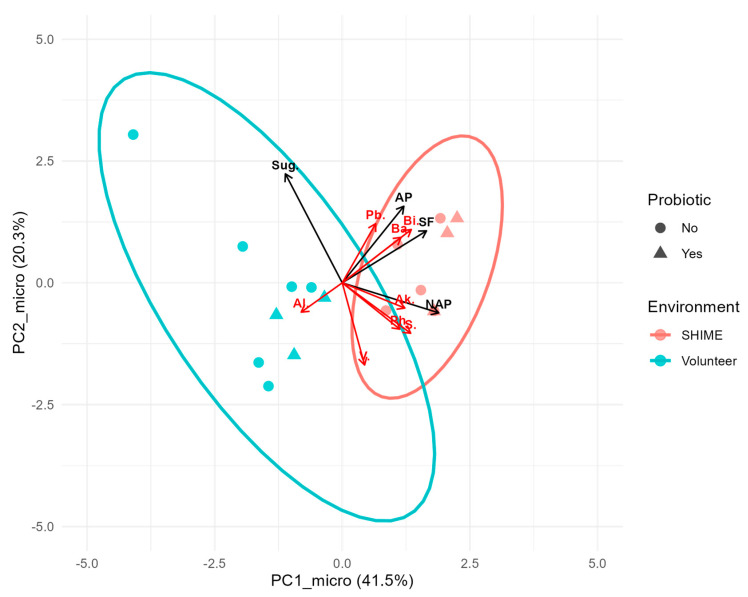
Biplot of principal components based on the abundance of the eight most prevalent genera shared between SHIME and fecal samples. Points, red vectors, and black vectors represent samples, rescaled genera loadings, and overlaid principal components of dietary macronutrients, respectively. Ellipses mark 95% confidence regions for each microbiota culturing environment. Abbreviations: Pb.—*Parabacteroides*, Ph.—*Phascolarcobacterium*, Ba.—*Bacteroides*, Bi.—*Bifidobacterium*, V.—*Victivallis*, Al.—*Alistipes*, Ak.—*Akkermansia*, S.—*Sutterella*, Sug.—sugar, AP—animal protein, SF—Soluble fiber, and NAP- non-animal protein.

**Figure 7 nutrients-17-03093-f007:**
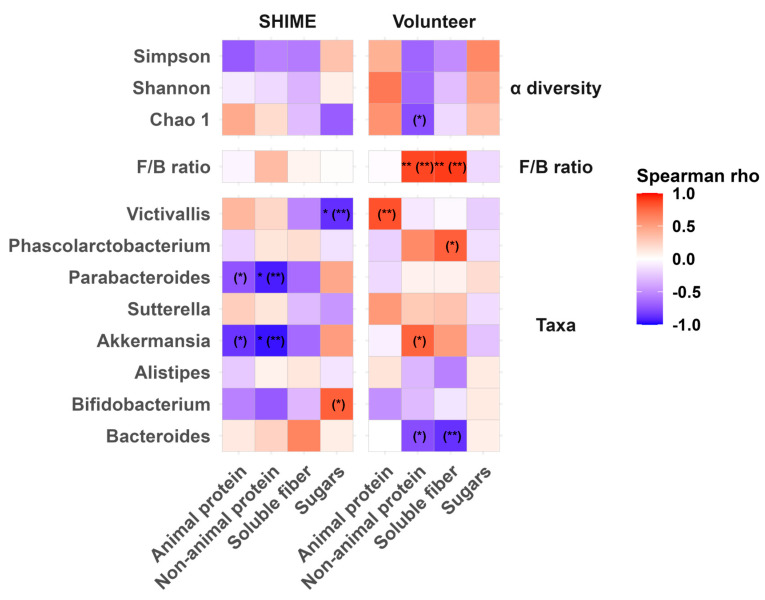
Heatmap of Spearman’s correlation coefficients between the optimal averaging window content of macronutrients in SHIME feed and food residues and the α-diversity, F/B ratio, and relative abundance of the most prevalent genera common for the microbiota populating SHIME and human feces. Statistically significant correlations after FDR correction were marked with ** < 0.01, and * < 0.05, and raw *p* values are in parentheses. FDR corrections were applied for each level of data (alpha diversity, F/B ratio, and 8 genera) separately.

**Table 1 nutrients-17-03093-t001:** Content of nutrients in standard and experimental diets for the volunteer and SHIME, mean ± standard deviation.

Type of Diet	Nutrient	Microbiota Cultivation Environment		SHIME’s Feed Content	
		Volunteer (g/Day)	SHIME (g/L)	Ingredient	Concentration (g/L)
Standard *	Animal protein	34.9 ± 23.7	1.0	Special peptone	1.0
	Non-animal protein	25.9 ± 13.5	2.0	Yeast extract	3.0
	Arabinogalactan + arabinoxylan	5.0 ± 2.8	1.7	Xylan	0.5
	Pectin	2.2 ± 1.6	2.0	Gum Arabic	1.2
	Resistant starch	1.7 ± 1.6	4.0	Pectin	2.0
	Soluble fiber	9.0 ± 3.3	7.7	Starch	4.0
	Sugars	77.0 ± 41.8	0.5	Glucose	0.4
Experimental	Animal protein	46.7 ± 13.1	3.8 ± 1.4	Special peptone	3.8 ± 1.4
	Non-animal protein	41.4 ± 6.5	3.4 ± 0.8	Yeast extract	5.4 ± 1.2
	Arabinogalactan + arabinoxylan	4.9 ± 1.3	2.3 ± 1.1	Xylan	0.7 ± 0.3
	Pectin	7.5 ± 1.6	3.2 ± 0.7	Gum Arabic	1.7 ± 0.8
	Resistant starch	5.1 ± 2.4	2.0 ± 0.8	Pectin	3.2 ± 0.7
	Soluble fiber	17.5 ± 3.3	7.6 ± 1.2	Starch	2.0 ± 0.8
	Sugars	66.4 ± 18.6	0.4 ± 0.1	Glucose	0.3 ± 0.1

* Standard diet—28-day mean intake of nutrients from the volunteer’s diary and standard SHIME feed medium nutrients/ingredients used during the 14-day stabilization period before the experiment; the SHIME media, besides indicated nutrients, contained 2 and 0.5 g of mucin and l-cysteine, respectively. Soluble fiber content is the sum of all fiber components indicated in the table.

## Data Availability

The full dataset that allowed for the generation of results reported in this study was archived in an open repository under: https://doi.org/10.5281/zenodo.14974398 (also shared in a previous publication) and under: https://doi.org/10.5281/zenodo.16751627 (exclusively for the current report).

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
