# Peer review of "A Comparison of the Response of the Human Intestinal Microbiota to Probiotic and Nutritional Interventions In Vitro and In Vivo—A Case Study"

_nutrients, 2025, doi:10.3390/nu17193093_

Round 1
Reviewer 1 Report
Comments and Suggestions for Authors
This is about “Comparison of the response of the human intestinal microbiota to probiotic and nutritional intervention in vitro and in vivo”
The study's premise of comparing in vitro and in vivo responses is interesting. However, the manuscript suffers from critical flaws that cannot be addressed with a simple revision.
Comment 1. The most significant flaw is the use of a single participant. This is a case study, not a comparative study. Without a cohort, the findings are anecdotal and lack statistical power. The results cannot be generalized to the broader population, and therefore, the core objective of "comparison" is not scientifically valid.
Comment 2. The manuscript fails to provide a valid ethics approval number from an Institutional Review Board (IRB) or a similar ethics committee. This is a mandatory requirement for any research involving human subjects.
Comment 3. A principle of scientific research is reproducibility. The study's design, based on a single sample, makes it impossible to replicate the in vivo findings. Furthermore, the lack of information on multiple replicates for the in vitro SHIME experiment prevents us from assessing the stability and reliability of your model.
Comment 4. The Materials and Methods section is missing critical information required for replication. The manuscript does not specify the volunteer's diet, the exact composition of the SHIME nutrient medium, or the total caloric intake for each. This lack of transparency is unacceptable.
Comment 5. With only one participant, no robust statistical analysis can be performed. The manuscript does not explain how the "differences" were quantified or assessed, leaving the conclusions unsupported by any rigorous evidence.
Comment 6. The final sentence of the introduction fails to articulate a clear hypothesis or the study's specific purpose. It does not effectively justify why the in vivo and in vitro samples were compared, making the entire premise of the study weak.
Comment 7. The abstract and conclusion overstate the findings. Given the single-participant design, it is inappropriate to draw general conclusions about the predictive validity of the SHIME model. The language must be revised to reflect the severe limitations of a case study.
Comment 8. The manuscript vaguely refers to "probiotic and nutritional interventions." The specific type, dose, and duration of these interventions are not clearly detailed. It is also unclear how the individual effects of the probiotic and the nutritional intervention were isolated.
Comment 9. Provide the change of probiotics' number in vitro and in vivo.
Author Response
Thank you very much for all your comments and for taking the time to read our manuscript. Please find a detailed response in the attached file.

|
Comment |
Reply |
|
This is about “Comparison of the response of the human intestinal microbiota to probiotic and nutritional intervention in vitro and in vivo” The study's premise of comparing in vitro and in vivo responses is interesting. However, the manuscript suffers from critical flaws that cannot be addressed with a simple revision. |
Thank you for this comment. Below, we have addressed the particular concerns that led to this general comment. |
|
Comment 1. The most significant flaw is the use of a single participant. This is a case study, not a comparative study. Without a cohort, the findings are anecdotal and lack statistical power. The results cannot be generalized to the broader population, and therefore, the core objective of "comparison" is not scientifically valid.
|
Thank you. We agree that the studies with many participants are methodologically valid when researchers want to look at the global effect of the intervention on the microbiota. Nevertheless, our methodological approach was designed to assess and understand whether we could model microbiota responses of a particular person, not a global response of a representative sample for a given population. To articulate this more clearly, we have added the following sentence at the end of the introduction: “In this study, dietary and LGG interventions were applied to SHIME system and the human donor in parallel to discern whether the behaviour of individual microbiota may be modelled in an in vitro setting.“
|
|
Comment 2. The manuscript fails to provide a valid ethics approval number from an Institutional Review Board (IRB) or a similar ethics committee. This is a mandatory requirement for any research involving human subjects. |
Thank you for this comment. The ethical approval was mentioned in the field under the Conclusion section under the heading “Institutional Review Board Statement: “. |
|
Comment 3. A principle of scientific research is reproducibility. The study's design, based on a single sample, makes it impossible to replicate the in vivo findings. Furthermore, the lack of information on multiple replicates for the in vitro SHIME experiment prevents us from assessing the stability and reliability of your model. |
Thank you for this comment. We agree that the study cannot be reproduced in the context of the structure of the microbiota, which is the case between individuals or even different time points within an individual. However, the study can be reproduced in terms of the main findings. If one uses our protocol for running a duplicate experiment, they should find that 1. The dynamic in vitro system does not allow the establishment of a microbiota that is compositionally similar to the original microbiota of the donor, and 2. Established microbiota react differently to applied interventions. |
|
Comment 4. The Materials and Methods section is missing critical information required for replication. The manuscript does not specify the volunteer's diet, the exact composition of the SHIME nutrient medium, or the total caloric intake for each. This lack of transparency is unacceptable. |
Thank you for this. We agree, but the wealth of data is too great to include in the manuscript. This is why we have provided access to open data deposited in the open repository of ZENODO. It includes all the details together with the composition of the media, food items that were consumed by the volunteer and their nutritional content. The link to the data is provided below the Conclusion section under the “Data Availability Statement”, In addition, we have now added citations of these resources in the methodology, along with the sentence “The mean intakes of dietary macronutrients and the nutritional compositions of SHIME feed media before and during the intervention were summarised in Table 1. Daily detailed nutrition data, including information about the meals catered to volunteer, were deposited in an open data resource [21,22].” |
|
Comment 5. With only one participant, no robust statistical analysis can be performed. The manuscript does not explain how the "differences" were quantified or assessed, leaving the conclusions unsupported by any rigorous evidence. |
We appreciate this comment. To address it, we have now emphasized in the title, abstract and conclusion that the study was based on a single participant and the evidence gathered is “anecdotal”. The differences were assessed based on the correlation coefficients, given that the microbiota from each culturing environment was taken multiple times (10) during the interventions and at least once before the interventions. In addition, a global test- a linear model with type III ANOVA was now used, explaining clearly that interventions caused different effects on microbiota structure in both environments (following sentence “ (...) a significant interaction between Environment and PC1_diet and Environment and Probiotic on PC1_micro (~7% and 4% variance explained, respectively) was identified.” |
|
Comment 6. The final sentence of the introduction fails to articulate a clear hypothesis or the study's specific purpose. It does not effectively justify why the in vivo and in vitro samples were compared, making the entire premise of the study weak.
|
Thank you for this comment. We have now changed the wording to more clearly articulate our aims and the knowledge gaps we wanted to address. The final part of the introduction now states,” In this study, dietary and LGG interventions were applied to SHIME system and the human donor in parallel to discern whether the behaviour of individual microbiota may be modelled in an in vitro setting. To our knowledge, this is the first study that attempted a comparison of the microbiota response to dietary and probiotic intervention in humans and simulators in parallel.” |
|
Comment 7. The abstract and conclusion overstate the findings. Given the single-participant design, it is inappropriate to draw general conclusions about the predictive validity of the SHIME model. The language must be revised to reflect the severe limitations of a case study. |
Thank you for this comment. We have now addressed it by the following changes in the mentioned fragments: Title: words “-a case study” were added to the title Abstract: The conclusion part was changed, and now it reads ”Conclusions: The anecdotal evidence presented in this study suggested that current in vitro technology enables the reproduction of some of the microbiota responses that are well-known from in vivo research. However, further work is required to enable simulations of an individual microbiota.” Conclusions: The following sentences were added: “Although this study included no replication and only a single donor, to our knowledge, it presents the first attempt to compare the response of an individual microbiota to probiotic and dietary intervention in the donor and SHIME in parallel. We propose that the alteration of the microbial community during the establishment in the in vitro setting, observed here as well as in previous studies, is one of the several reasons contributing to the divergence of individual microbiota behaviour between artificial and natural environments.” |
|
Comment 8. The manuscript vaguely refers to "probiotic and nutritional interventions." The specific type, dose, and duration of these interventions are not clearly detailed. It is also unclear how the individual effects of the probiotic and the nutritional intervention were isolated. |
Thank you. You are absolutely right. Initially, we thought that the reference to a previous publication that contained the methodology would be adequate. We have now expanded the methodology section. Regarding separating the effect of probiotics, since it was negligible, we did not correct for it in the initial statistics, but now we have revised it to accommodate this comment and included a linear model with type III ANOVA as our main statistics. This statistical test considers all variables at once, including the parallelism of interventions. |
|
Comment 9. Provide the change of probiotics' number in vitro and in vivo. |
Thank you for the comment. The study was not designed to quantify specifically LGG, but to compare its impact on the microbiota in SHIME and in the volunteer. |
|
Thank you for all your kind comments and for taking the time to read our manuscript. |
|
Reviewer 2 Report
Comments and Suggestions for Authors
The authors investigated the responses of individual human gut microbiota to dietary and probiotic interventions both in vivo and in vitro using the SHIME system. The integration of high-throughput sequencing, careful dietary control, and correlation analyses provides valuable insights into microbiota-diet interactions and the limitations of in vitro modelling. This work represents a meaningful contribution to the field of nutritional microbiomics and precision nutrition. Before the manuscript can be considered for publication, some revisions are necessary to improve clarity, readability, and accuracy.
Minor revisions:
- Revise grammar in introduction, especially lines 46, 51 and 53-54.
- Revise typos
- Revise English use in some expressions like in lines 96 and 387.
Introduction:
The data displayed is quite dense, making it challenging for readers to follow the logical progression from disease context to the rationale for using SHIME. Consider tightening sentences and grouping related concepts more cohesively.
The sifts between gut microbiota, diet, precision nutrition, probiotics, and in vitro models are somewhat abrupt. The narrative could benefit from smoother transitions linking diet and probiotics to the need for in vitro models.
Also, emphasize more clearly ‘how combined dietary and probiotic interventions affect microbiota in SHIME versus in vivo’ as the central research question could improve the general contribution of the paper.
Condense some concepts. For instance, the role of diet and probiotics in lines 52-66 and 88-95.
Materials and methods:
Please, provide SHIME adjustments to mimic the volunteer’s diet. The rationale for specific percentage changes could be more explicitly justified.
Figure 2 mentioned the sampling scheme, but the text could briefly summarize the timing and rationale for sampling rather than relaying on the figure alone. This will improve narrative.
DNA extraction is described in triplicate for faecal samples and once per SHIME compartment. What is the reasoning for this difference?
Results:
Multiple references to supporting information (Figures S1-S2 and Tables S1-S10) are made. While informative, the text relies heavily on these. A brief summary of key trends from the supplementary material in the main text would improve readability.
While indices are reported (Chao1, Shannon, Simpson, F/B ratio), interpretation could be more concise. The current text mixes results and potential explanations, which may be more appropriate for the Discussion.
Discussion:
Some parts repeat data already presented in the results. This could be condensed and focused more on interpretation and contextualization.
The discussion explains discrepancies between SHIME and fecal microbiota, but at times the argument is complex and could benefit from clearer, concise statements summarizing the key mechanisms (e.g., stochasticity, coadaptation, feed composition).
While the potential of SHIME for precision nutrition is mentioned, the discussion could benefit from a concise paragraph explicitly stating how the findings can inform future in vitro studies or nutritional interventions.
Conclusions:
The conclusions summarize the study well, but statements such as “the system enabled microbiota responses that were known from multiple earlier human and animal studies” are vague. It would be stronger to briefly specify which responses were accurately reproduced.
The manuscript mentions that SHIME could help in precision nutrition or personalized medical strategies. This is forward-looking, but the authors should clarify that, in this study, SHIME did not faithfully replicate individual microbiota responses.
The conclusion briefly mentions “further advances in in vitro technology,” but it could be more precise by suggesting what specific improvements (e.g., stabilization procedures, feed formulation, inoculum handling) are most critical.
With these changes, I believe the manuscript would be up for acceptance to be published.

Author Response
Thank you very much for all your comments and for taking the time to read our manuscript. Please find a detailed response in the attached file.

|
No. |
Comment |
Reply |
|
1. |
The authors investigated the responses of individual human gut microbiota to dietary and probiotic interventions both in vivo and in vitro using the SHIME system. The integration of high-throughput sequencing, careful dietary control, and correlation analyses provides valuable insights into microbiota-diet interactions and the limitations of in vitro modelling. This work represents a meaningful contribution to the field of nutritional microbiomics and precision nutrition. Before the manuscript can be considered for publication, some revisions are necessary to improve clarity, readability, and accuracy. Minor revisions:
|
Thank you very much for this comment. We have now revised the grammar and the use of language in the parts identified in this comment. We have also screened for typos. |
|
2. |
Introduction: The data displayed is quite dense, making it challenging for readers to follow the logical progression from disease context to the rationale for using SHIME. Consider tightening sentences and grouping related concepts more cohesively. The sifts between gut microbiota, diet, precision nutrition, probiotics, and in vitro models are somewhat abrupt. The narrative could benefit from smoother transitions linking diet and probiotics to the need for in vitro models. Also, emphasize more clearly ‘how combined dietary and probiotic interventions affect microbiota in SHIME versus in vivo’ as the central research question could improve the general contribution of the paper. Condense some concepts. For instance, the role of diet and probiotics in lines 52-66 and 88-95. |
Thank you very much. Now we have reorganised the text in the introduction, added some explanatory notes to make it more coherent, but overall shortened the text by removing some repetitions. Text that was added/changed was highlighted in yellow. |
|
3. |
Materials and methods: 1. Please, provide SHIME adjustments to mimic the volunteer’s diet. The rationale for specific percentage changes could be more explicitly justified. 2. Figure 2 mentioned the sampling scheme, but the text could briefly summarize the timing and rationale for sampling rather than relaying on the figure alone. This will improve narrative. 3. DNA extraction is described in triplicate for faecal samples and once per SHIME compartment. What is the reasoning for this difference?
|
Thank you for pointing this out. We have added the following changes to this section: 1. A detailed explanation in section 2.2 (highlighted text) and changed Figure 1 to Table 1. 2. Sampling days in the sentence “Whereas during the intervention, sampling was performed 10 times, of which 5 were during the combined dietary and probiotic (on experimental days 3, 6, 8, 10, 12) and the remaining during probiotic washout and continued dietary intervention (on experi-mental days 16, 18, 22, 25, 28).” were articulated. 3. The following explanation: “The sampling scheme relied on the volunteer’s availability to provide a fresh fecal sample on-site. The volunteer was instructed on the required number of samples for each study stage, with the condition that successive samples be collected at least one day apart.” was added. 4. We have now clarified: “To reveal the structure of microbial community a high-throughput sequencing of the V3-V4 regions of 16S rRNA was performed using DNA extracts pooled from three replicates taken for each sample.” |
|
4. |
Results: Multiple references to supporting information (Figures S1-S2 and Tables S1-S10) are made. While informative, the text relies heavily on these. A brief summary of key trends from the supplementary material in the main text would improve readability. While indices are reported (Chao1, Shannon, Simpson, F/B ratio), interpretation could be more concise. The current text mixes results and potential explanations, which may be more appropriate for the Discussion. |
Thank you, we have now removed references to tables S1-10 and instead run a sensitivity analysis, which was based on these and additional simulated data summarised in Figure 5. Key trends of F/B ratio behaviour (Figure S1 and 2) were mentioned in the appropriate parts of the results and discussion sections. Detailed data were included in the open repository (referenced under data availability statement). We have also removed the text discussing the diversity indices from the Results section. |
|
5. |
Discussion: Some parts repeat data already presented in the results. This could be condensed and focused more on interpretation and contextualization. The discussion explains discrepancies between SHIME and fecal microbiota, but at times the argument is complex and could benefit from clearer, concise statements summarizing the key mechanisms (e.g., stochasticity, coadaptation, feed composition). While the potential of SHIME for precision nutrition is mentioned, the discussion could benefit from a concise paragraph explicitly stating how the findings can inform future in vitro studies or nutritional interventions. |
Thank you very much for this comment. We have now made changes across the discussion to reduce inappropriate result repetitions, discussed key mechanisms under separate paragraphs and added a new paragraph at the end of the section “Summarizing, findings of this and other studies suggest that standard protocols used for running in vitro experiments do not preserve the original structure of human microbiota. This likely contributes to shifts in the microbiota responses to interventions. Further studies should hence explore modifications to standard protocols aimed at the preservation of an individual microbiota structure. Such improvements could enable the use of in vitro technology to support developments in personalized health management strategies, including precision nutrition.“ |
|
6. |
Conclusions: 1. The conclusions summarize the study well, but statements such as “the system enabled microbiota responses that were known from multiple earlier human and animal studies” are vague. It would be stronger to briefly specify which responses were accurately reproduced. 2. The manuscript mentions that SHIME could help in precision nutrition or personalized medical strategies. This is forward-looking, but the authors should clarify that, in this study, SHIME did not faithfully replicate individual microbiota responses. 3. The conclusion briefly mentions “further advances in in vitro technology,” but it could be more precise by suggesting what specific improvements (e.g., stabilization procedures, feed formulation, inoculum handling) are most critical. With these changes, I believe the manuscript would be up for acceptance to be published. |
1. Thank you very much, we have now added the following statement to the conclusion section: “Particularly, we observed a positive correlation of Bifidobacterium abundance to the increase of glucose in the feed medium and the enrichment of the former Lactobacillus genus during probiotic supplementation.” 2. We have added the following paragraph to the Conclusions “Although this study included no replication and only a single donor, to our knowledge, it presents the first attempt to compare the response of an individual microbiota to probiotic and dietary intervention in the donor and SHIME in parallel. We propose that the alteration of the microbial community during the establishment in the in vitro setting, observed here as well as in previous studies, is one of the several reasons contributing to the divergence of individual microbiota behaviour between artificial and natural environments.” 3. We agree and we have modified the following sentence in the conclusions “Further advances in the in vitro technology that would be aimed at the preservation of the structure of microbial communities contained in the inoculum, e.g. preparing frozen rather than fresh inocula, and adjusting the feed media composition, are essential to enhance the predictive power of such systems.” |
|
Thank you for all your kind comments and for taking the time to read our manuscript. |
||
Reviewer 3 Report
Comments and Suggestions for Authors
nutrients-3846056-peer-review-v1
This is an interesting research project, comparing results from probiotic supplementation and effects on microbiota evaluated by artificial model and volunteer (in vitro and in vivo). Work is interesting and deserve attention form the Editor, however, some adjustments and corrections will need to be taken into account by the authors.
Abstract: Ln29-31: Maybe this part will be more appropriate to be moved to the Methods section.
Maybe Results section in the abstract can be enriched with some more specific data?
Introduction is presented with sufficient details and provided with general information regarding further investigated points. Authors have clearly pointed out benefits and limitations in using SHIME models and questions regarding appropriate experimental set-ups, where details on experimental set-ups pays an important role in interpretation of the results.
Objective of the study was clearly stated.
To some extent, material and methods are combined with results. Please, try to be more correct and clearly separate the material and methods and applied procedures from the results and discussion.
On Figure 1, applied diet image looks like Asian food. Is this random image or was it a real food diet for participating volunteers? Maybe cartoon images will be more appropriate?
On figure 2: Please, replace cartoon for the LGG. Lactobacilli normally are not motile and do not have flagella.
In the text it will be appropriate to state how LGG was prepared and mentioned cell number confirmed. This was lyophilized commercial preparation. Or was suspension prepared at laboratory? These details are important and needs to be provided.
Title of the figures needs to be corrected and enriched. Title needs to be self-explicative.
Section 2.4. needs to be presented with more details. AS well, applied material and further DNA manipulation needs to be described with sufficient details. Maybe this can be provided as supplementary material (depends on the choice from authors and recommendations from Editor and journal regulations), but this information is important. Moreover, for all materials and equipment’s, authors will need to provide their suppliers. This includes name of the company and appropriate address. Address needs to include city, state (in case of the federal country) in abbreviated way, and name of the country. In following occasions, only name of the company will be sufficient. Please, try to use headquarter address of the company and not local distributors.
Results needs to be presented with more details. As example, on Ln174, Please, start with kind of introduction sentence and then present your results..
Maybe it will be interesting if authors can point to the differences in the microbiota between model and volunteers with a bit more attention. Form this point, maybe providing some critics to the applied models can be interesting point that will need more attention form the authors.
Fact that lactobacilli were not really recorded and fact that LGG was applied and was expected to be recorded in the analysis merits really attention. Authors have provided some comments regarding this fact, however, maybe a bit more extended discussion on this point will be beneficial to be provided.
Other point is fact that authors are working with DNA, and in such as analysis DNA even form dead cells can be recorded and give some results. When we talk about probiotics, we focus on life bacteria. However, how can this point be discussed? Authors have performed any additional step in their work that will be predominantly isolating DNA only from life bacterial population. Please provide details on this issue. If you have had isolated total DNA (from life and dead cells) how can this interfere with you results? This issue merits a paragraph in the discussion section and some arguments regarding applied procedures and defending your experimental approaches and applied methods that needs to be provided.
Results are presented well and with sufficient details and appropriate visual/illustrative material.
Discussion is analysis the obtained results and comparing with available literature.
Ln480, 484, etc: Please, some references needs to be adjusted to the requirements form the journal and publisher. Change from Devarakonda et al. (2024) to Devarakonda et al. [83]. Etc.
Author Response
Thank you very much for all your comments and for taking the time to read our manuscript. Please find a detailed response in the attached file.

|
No. |
Comment |
Answer |
|
1. |
This is an interesting research project, comparing results from probiotic supplementation and effects on microbiota evaluated by artificial model and volunteer (in vitro and in vivo). Work is interesting and deserve attention form the Editor, however, some adjustments and corrections will need to be taken into account by the authors. |
Thank you very much. We appreciate your insights. |
|
2. |
Abstract: Ln29-31: Maybe this part will be more appropriate to be moved to the Methods section. Maybe Results section in the abstract can be enriched with some more specific data? |
Thank you very much. We agree. We have now modified the abstract by moving the methodology part as suggested, and included the following description under the results part “Analysis of 17 samples revealed that predominantly diet and to a lesser extent probiotic had a divergent effect to microbiota’s fluctuations dependent on the culture environment.” |
|
3. |
Introduction is presented with sufficient details and provided with general information regarding further investigated points. Authors have clearly pointed out benefits and limitations in using SHIME models and questions regarding appropriate experimental set-ups, where details on experimental set-ups pays an important role in interpretation of the results. Objective of the study was clearly stated. |
Thank you for sharing your positive opinion. We have modified the introduction only slightly (more stylistically than content-wise) to accommodate the comments of other reviewers. |
|
4. |
To some extent, material and methods are combined with results. Please, try to be more correct and clearly separate the material and methods and applied procedures from the results and discussion. |
Thank you very much for pointing this out. Indeed, there was some text describing dietary intervention and how it changed in comparison to the standard diet in the methodology section 2.2, but now we have removed it. |
|
5. |
On Figure 1, applied diet image looks like Asian food. Is this random image or was it a real food diet for participating volunteers? Maybe cartoon images will be more appropriate? |
As for the visual form of the figure, we truly appreciate your suggestion and fully understand it. Initially, we decided to include a real photo to emphasize the authenticity of the intervention and give readers a more tangible sense of what such a diet looks like in practice. On the other hand, we have been asked to describe dietary intervention in a more detailed way and included Table 1 that now replaces Figure 1. |
|
6. |
On figure 2: Please, replace cartoon for the LGG. Lactobacilli normally are not motile and do not have flagella. |
Thank you very much, you are right. To address this comment, we have replaced the bacterium with a pill to visualise intervention in a more precise way. |
|
7. |
In the text it will be appropriate to state how LGG was prepared and mentioned cell number confirmed. This was lyophilized commercial preparation. Or was suspension prepared at laboratory? These details are important and needs to be provided. Title of the figures needs to be corrected and enriched. Title needs to be self-explicative. |
To address the first part of this comment, we have now added the following paragraph to section 2.2 “A commercially available LGG supplement in the form of capsules, claimed to contain 6 x 109 LGG cells per capsule was used in this study. Both the volunteer and the SHIME ingested the supplement twice a day. The volunteer consumed the capsules, while SHIME was inoculated with a small volume of sterile water in which the contents of 3 capsules were suspended (1.5 capsules per each stomach/ileum vessel). SHIME’s stomach was administered LGG right after filling with the fresh feed medium at pH 2; thereby exposing the probiotic to simulated GIT conditions from the stomach to the colon.” We have also updated figure captions. All changes were highlighted in yellow. |
|
8. |
Section 2.4. needs to be presented with more details. AS well, applied material and further DNA manipulation needs to be described with sufficient details. Maybe this can be provided as supplementary material (depends on the choice from authors and recommendations from Editor and journal regulations), but this information is important. Moreover, for all materials and equipment’s, authors will need to provide their suppliers. This includes name of the company and appropriate address. Address needs to include city, state (in case of the federal country) in abbreviated way, and name of the country. In following occasions, only name of the company will be sufficient. Please, try to use headquarter address of the company and not local distributors. |
We agree with this comment. These details are all necessary to reproduce the outcomes. However, through the methodology section, we have used a reference that includes all of the requested information. Now, in section 2.1, we have clarified that “This previous publication reported a subset of data from the same experiment; hence, to avoid repetition, methodological details contained in the current work were limited to a minimum.” |
|
9. |
Results needs to be presented with more details. As example, on Ln174, Please, start with kind of introduction sentence and then present your results.. |
Thank you for pointing this out. We have now made substantial changes to the result section, which are all highlighted in yellow. |
|
10. |
Maybe it will be interesting if authors can point to the differences in the microbiota between model and volunteers with a bit more attention. Form this point, maybe providing some critics to the applied models can be interesting point that will need more attention form the authors. Fact that lactobacilli were not really recorded and fact that LGG was applied and was expected to be recorded in the analysis merits really attention. Authors have provided some comments regarding this fact, however, maybe a bit more extended discussion on this point will be beneficial to be provided. |
Thank you for these insights. We have now changed the main statistics (to a linear model with type III ANOVA), which nicely demonstrates general differences between SHIME and the volunteer in microbiota composition and highlights that each microbiota responded in a different way to the intervention. We have also included the following text in the discussion “In contrast to the human gastrointestinal tract, the SHIME model provides stable and controlled conditions that facilitate the persistence and proliferation of supplemented strains, whereas in vivo their colonization is often limited by microbial competition, rapid intestinal transit, and host immune responses. Moreover, colonization by Lactobacillus is frequently transient, meaning that their presence in feces may be sporadic and below the detection threshold of 16S rRNA sequencing. In addition, the 16S rRNA may suffer from primer-dependent amplification bias towards low-abundance DNA, which ultimately can result in false negatives. Moreover, LGG may transiently adhere to the mucosal surface of the small intestine or proximal colon rather than being shed in feces, which limits its detectability in stool-based sequencing.” |
|
11. |
Other point is fact that authors are working with DNA, and in such as analysis DNA even form dead cells can be recorded and give some results. When we talk about probiotics, we focus on life bacteria. However, how can this point be discussed? Authors have performed any additional step in their work that will be predominantly isolating DNA only from the bacterial population. Please provide details on this issue. If you have had isolated total DNA (from life and dead cells) how can this interfere with you results? This issue merits a paragraph in the discussion section and some arguments regarding applied procedures and defending your experimental approaches and applied methods that needs to be provided. |
Thank you for this. Yes, you are very right. In this study, we did not use any means of removing dead cell DNA. There are two reasons for this: 1. This is a standard approach taken by the majority of similar studies, 2. Both SHIME and Volunteer excreted waste from their respective colons, which meant that dead cells would be partly removed during each excretion. Furthermore, fresh feed that was provided nourished live microbiota, which would then overpower the abundance of dead cells. This may justify the broad use of the selected approach. Nevertheless, we would like to emphasise here that distinguishing between live and dead microorganisms was not the aim of this study. Our main purpose was to compare the response of the microbiota in SHIME and the Volunteer. |
|
12. |
Results are presented well and with sufficient details and appropriate visual/illustrative material. Discussion is analysis the obtained results and comparing with available literature. Ln480, 484, etc: Please, some references needs to be adjusted to the requirements form the journal and publisher. Change from Devarakonda et al. (2024) to Devarakonda et al. [83]. Etc. |
Thank you very much for your suggestions and kind comments. We have included suggested changes to the references across the manuscript. |
Reviewer 4 Report
Comments and Suggestions for Authors
The manuscript entitled “Comparison of In Vivo and SHIME®-Simulated In Vitro Gut Microbiota Response to Diet and Probiotic Intervention in the Same Donor” directly addresses the issue of reproducibility of in vitro models with respect to the actual response of the human gut microbiota. Its originality lies in performing the comparison within the same donor, thereby reducing interindividual variability and allowing a more precise evaluation of how reliably the SHIME model can reflect real dietary and probiotic changes. The findings contribute to the ongoing debate on the use of dynamic simulated systems to anticipate personalized responses and reduce the reliance on lengthy and costly clinical trials.
However, several points could be improved to enhance the quality of the manuscript.

Comments on the Quality of English Language
The English could be improved to more clearly express the research
Author Response
Thank you very much for all your comments and for taking the time to read our manuscript. Please find a detailed response in the attached file.

|
No. |
Comment |
Answer |
|
1. |
The manuscript entitled “Comparison of In Vivo and SHIME®-Simulated In Vitro Gut Microbiota Response to Diet and Probiotic Intervention in the Same Donor” directly addresses the issue of reproducibility of in vitro models with respect to the actual response of the human gut microbiota. Its originality lies in performing the comparison within the same donor, thereby reducing interindividual variability and allowing a more precise evaluation of how reliably the SHIME model can reflect real dietary and probiotic changes. The findings contribute to the ongoing debate on the use of dynamic simulated systems to anticipate personalized responses and reduce the reliance on lengthy and costly clinical trials. However, several points could be improved to enhance the quality of the manuscript.
|
Thank you for taking the time to read our manuscript and sharing valuable comments. |
|
2. |
• Avoid using independent-sample t-tests on α-diversity indices obtained from longitudinal repeated measures on the same subject and system: the assumption of independence is violated and normality is questionable. Mixed-effects models or nonparametric tests for repeated measures (e.g., LMM on α-diversity or on CLR-transformed data) would be more appropriate. The text currently reports “independent sample t-test (p<0.05)” for K vs. L differences. |
Thank you for this comment. We have now replaced the t-test with Wilcoxon’s signed-rank test, treating K and L samples taken at the same time point as pairs. We did not, however, apply statistics that would consider repeated measures since we had only one replicate per time-point for each sample (DNA was extracted in triplicate but for NGS replicates were pooled). We have also revised the remaining statistics. As our main test, we have selected LM with type III ANOVA on the CLR transformed abundances. Spearman’s correlations were retained to explain effects found by the main test in more detail, but we have applied FDR as advised. All statistical tests and data transformations were described under section 2.5, and changes were highlighted. |
|
3. |
• Clarify a priori what the primary endpoint was (e.g., distance between in vitro vs. in vivo profiles, variation in shared genera, Firmicutes/Bacteroidetes ratio, LGG detection) and what the secondary endpoints were. At present, the narrative alternates between α-diversity, F/B, PCoA, and multiple correlations without a clearly defined primary outcome, which risks data dredging. |
We appreciate this comment, and to address it, we have removed Figure 8, which was a heat map of correlations between the abundance of genera highly prevalent in both SHIME and volunteer. Furthermore, we have now added the following explanations: In section 2.5 (methods)- “Perform linear modelling as the main analysis to compare the response of the microbiota to interventions in two different environments” In section 3.3 (results)- “To quantitatively assess whether the probiotic and dietary intervention had different outcomes in two microbiota culturing environments, a linear model was fitted to bacterial abundance and intervention data” and “Since the microbiota composition was significantly affected by the interaction between the Environment and PC1_diet, a separate analysis investigating responses to macronutrients in each environment was performed.” |
|
4. |
• For β-diversity, the use of Jaccard (presence/absence) is inconsistent with the statement “all compositional data were CLR-transformed prior to analysis.” For CLR data, the natural distance is the Aitchison (Euclidean). The PCoA/ordinations should be redone using Aitchison (or Bray–Curtis on relative abundances), followed by PERMANOVA with stratification by subject/phase and verification of β-dispersion. The manuscript reports Jaccard for PCoA despite declaring CLR transformation. |
Thank you for pointing this out. Yes, you are absolutely right. The issue was in the description of the methodology, not in the actual test carried out. The methodology section was now revised, and data analysis was described in detail in section 2.5. |
|
5. |
• The numerous Spearman correlations (genera × nutrients × time windows) require multiple testing correction (FDR), which is not reported. Moreover, given the temporal autocorrelation and repeated measures, repeated-measures correlation (rmcorr) or mixed models with lag (2 14 days windows) would be preferable. |
Thank you for this comment. Indeed, we have not used the FDR correction initially. Now we have revised our statistical approach, and multiple comparisons were corrected either using FDR or Bonferroni adjustment (specified in the results section with the reported results). |
|
6. |
• Regarding the probiotic, key details are missing: brand/product, strain (ATCC 53103), CFU/day, mode of administration, and participant adherence. In SHIME, specify the inoculation point (stomach? proximal colon? timing), frequency, and viability checks. The absence of Lactobacillus in fecal samples during and after LGG supplementation contradicts expectations; strain-specific qPCR/targeted methods should be considered to draw conclusions on colonization failure. |
Thank you for these detailed comments. We have now included the following text in the first and second paragraph of section 2.2 “Both the SHIME model and the volunteer were subjected to parallel interventions: a 28-day dietary modification and supplementation with a commercial formulation containing LGG (ATCC 53103) administered during the first 14 days of dietary inter-vention. A commercially available LGG supplement in the form of capsules, claimed to contain 6 x 109 LGG cells per capsule, was used in this study. Both the volunteer and the SHIME ingested the supplement twice a day. The volunteer consumed the capsules, while SHIME was inoculated with a small volume of sterile water in which the contents of 3 capsules were suspended (1.5 capsules in each stomach/ileum vessel). SHIME’s stomach was administered LGG right after filling with the fresh feed medium at pH 2; thereby exposing the probiotic to simulated GIT conditions from the stomach to the colon. “
We did not focus on the viability of the probiotic. Instead, we were interested in what impact the probiotic administration had on the overall microbiota as per NGS analysis. We have emphasized this at the end of the introduction section in the following sentence: “In this study, dietary and LGG interventions were applied to SHIME system and the human donor in parallel to discern whether the behaviour of individual microbiota may be modelled in an in vitro setting.” |
|
7. |
• The SHIME feed does not include fats (“but no fat”), while in the volunteer’s diet fat was reduced but not eliminated. This discrepancy may alter microbial physiology via bile acids and lipid/emulsion dynamics, making in vitro vs. in vivo comparisons less reliable. This needs further discussion, and if possible, inclusion of a bile/fat modulator or a stronger justification for the exclusion. |
This is a very valid point. We have now included the following explanation in the second paragraph of section 2.2: “Fat was not included, as in this study, we chose to modify a standard SHIME feed by an exclusive adjustment of nutrient proportions. Nevertheless, the diet of the volunteer was not fat-free.” This choice was dictated by the fact that the studies leading to the development of SHIME and the manufacturer’s protocol claim that the medium represents food residues. The same medium is used across multiple studies now, and hence we did not want to change its ingredients, only their proportions. However, pancreatic juice (which also contains bile acids/salts) was added to SHIME during the whole experimental period as recommended in the system manufacturer’s protocol and as used in multiple published works. |
|
8. |
• Clarify whether the three fecal DNA extracts were pooled (the text refers to “composite samples of DNA extracts”) and why extraction was performed “once for each of the six distal colon vessels” (resulting in less technical replication in SHIME). Discuss the impact of pooling on variance and the ability to test differences. |
Thank you for this comment. We were not aware that this part was not clear, but defining technical replication is very important indeed. We did not have any technical replicates in sequencing. Pooling extracts was used to make sure that the samples were representative, since a very small part of each sample was used for extraction. Now we have added an underscored part to a sentence in section 2.4- “To reveal the structure of microbial community a high-throughput sequencing of the V3-V4 regions of 16S rRNA was performed using DNA extracts pooled from three replicates taken for each sample.”. Further detailed information about the sample preparation was included in our previous publication referenced across the methodology. This previous publication reported a subset of data from the study described in the proposed manuscript. |
|
9. |
• At one point you state that the strongest positive correlations occurred with an 8-day window (feces) and 14-day window (SHIME), but for subsequent analyses you use only 8 days; this needs to be justified, standardized, or both results presented as sensitivity analysis. |
Thank you for this comment. Initially, we have found that the abundance of a greater number of taxa correlates positively with the residues of some of the macronutrients when we use an 8-day averaging window for the residue content for feces. However, back then, we ran analyses using 1,2,4,8, and 14-day averaging windows. Now we have performed it with 1-day intervals up to 14 days to visualise any trends more explicitly (added Figure 5). This kind of sensitivity analysis revealed that, in fact, the optimal averaging window for the volunteer was 9 days and for SHIME 12 days. Although initially, we wanted to use the same averaging window for both microbiota cultivation environments, there was no strong rationale behind this choice other than the averaging window being the same for both microbiota culturing environments. Therefore, following your suggestion, we have now revised the initial approach and used an optimal averaging window for each setting instead. |
|
10. |
• The conclusions state that the study “confirms the value of SHIME as a reliable platform,” yet earlier you acknowledge the failure to simulate individual behavior. The tone should be more cautious, clearly distinguishing general predictive patterns from personalized replication. |
Thank you very much for this comment. We have now removed this sentence and emphasized particular responses of human microbiota that were replicated instead. “Particularly, we observed a positive correlation of Bifidobacterium abundance to the in-crease of glucose in the feed medium and the enrichment of the former Lactobacillus genus during probiotic supplementation.”. |
|
11. |
Section-specific comments • Abstract: Include key quantitative data (number of samples, main effect from PCoA/Adonis, number of significantly different shared genera) and the primary endpoint; currently it is more narrative. • Introduction: Explicitly highlight the research gap, i.e., “parallelism within the same donor for both macronutrient and probiotic intervention,” and how your study addresses it. • Discussion/Conclusions: Strengthen the discussion of the diet–feed mismatch (particularly fat/bile) as a plausible explanation for divergences. Reduce strong assertions about “confirmed value” and distinguish between qualitative validation and personalized prediction (which was not achieved). |
We are grateful for your insights regarding specific sections. We have now made several changes to the manuscript in accordance with your suggestions: ● Abstract- Results part now states “Analysis of 17 samples revealed that predominantly diet and to a lesser extent probiotics had a divergent effect on microbiota’s fluctuations dependent on the culture environment. Despite this, results from both in vitro and in vivo conditions aligned well with previously published data on the expected impact of dietary changes on the intestinal microbial community.” ● Introduction- the last paragraph of the introduction was now modified in the following way: “In this study, dietary and LGG interventions were applied to SHIME system and the human donor in parallel to discern whether the behaviour of individual microbiota may be modelled in an in vitro setting. To our knowledge, this is the first study that attempted a comparison of the microbiota response to dietary and probiotic intervention in human and simulator in parallel. “ ● Discussion- We have added the following sentences: “Furthermore, the lack of some food residues, such as fat and insoluble fibre, in the standard (and our experimental) SHIME feed may contribute to these shifts.” and also emphasised that the “SHIME feed was only matched with feed residues that could originate from the volunteer’s diet arbitrarily (…)” ● Conclusions- the statement was now removed and explanatory text added. |
Reviewer 5 Report
Comments and Suggestions for Authors
Rudzka et al. use the SHIME system to simulate the microbial communities in real samples and demonstrate its strong applicability. My comments are as follows:
- Line 112: Although the authors cite references, I recommend listing the detailed composition of the medium here.
- Figure 2: Should “6*10^9” be “6*10^9 CFU”?
- Figure 3: Since the p values for the Shannon and Simpson indices in panels b and c are greater than 0.05, these results could be moved to the supplementary materials rather than included in the main text.
- Line 187: Genus names should be italicized.
Author Response
Thank you very much for all your comments and for taking the time to read our manuscript. Please find a detailed response in the attached file.

|
No. |
Comment |
Answer |
|
1 |
Rudzka et al. use the SHIME system to simulate the microbial communities in real samples and demonstrate its strong applicability. My comments are as follows: Line 112: Although the authors cite references, I recommend listing the detailed composition of the medium here. |
We have now included Table 1 with details of nutrient intake and media composition as a replacement for the former Figure 1. |
|
2 |
Figure 2: Should “6*10^9” be “6*10^9 CFU”?
|
Thank you for pointing this out. We have now included CFU in the figure. |
|
3 |
Figure 3: Since the p values for the Shannon and Simpson indices in panels b and c are greater than 0.05, these results could be moved to the supplementary materials rather than included in the main text.
|
We appreciate this comment. We have made substantial changes to the statistics and data presentation. |
|
4 |
Line 187: Genus names should be italicized.
|
Thank you very much for pointing this out. We have found that in many places, genus names were not italicized, indeed. This was likely an issue with formatting during the manuscript upload. We have now italicized all names for genera. |
|
Thank you very much for your comments and for taking the time to read our manuscript. |
||
Round 2
Reviewer 1 Report
Comments and Suggestions for Authors
The statistical analysis that was problematic last time has been well organized. I hope more related research will be reported going forward.
Reviewer 2 Report
Comments and Suggestions for Authors
The authors have addressed the comments regarding the quality of their work and thus, the manuscript content has improved.
In this sense, I accept the manuscript as it is.
Reviewer 4 Report
Comments and Suggestions for Authors
The authors responded comprehensively to the reviewers' comments.
Comments on the Quality of English Language
The English could be improved to more clearly express the research.
Reviewer 5 Report
Comments and Suggestions for Authors
Ok, I think the author has addressed all my concerns.